# Distribution and Utilization of Vitamin E in Different Organs of Wild Bats from Different Food Groups

**DOI:** 10.3390/life14020266

**Published:** 2024-02-17

**Authors:** Diego Antonio Mena Canata, Mara Silveira Benfato, Francielly Dias Pereira, María João Ramos Pereira, Pabulo Henrique Rampelotto

**Affiliations:** 1Biophysics Department, Universidade Federal do Rio Grande do Sul, Porto Alegre 91501-970, RS, Brazil; 2Graduate Program in Cellular and Molecular Biology, Universidade Federal do Rio Grande do Sul, Porto Alegre 91501-970, RS, Brazil; 3Graduate Program in Animal Biology, Universidade Federal do Rio Grande do Sul, Porto Alegre 91501-970, RS, Brazil; 4Bioinformatics and Biostatistics Core Facility, Instituto de Ciências Básicas da Saúde, Universidade Federal do Rio Grande do Sul, Porto Alegre 91501-907, RS, Brazil

**Keywords:** alfa tocopherol, frugivorous, insectivorous, nectarivorous, vitamins, vampire bat

## Abstract

In this work, we examined the levels of vitamin E in the heart, liver, and kidneys of four species of adult male bats with distinct feeding habits. Our results indicate consistent vitamin E levels in the heart across all four bat species, suggesting the presence of regulatory mechanisms. Additionally, the liver displayed notably higher vitamin E levels in nectarivorous and frugivorous bats, while hematophagous bats exhibited lower levels, indicating a link between dietary intake and liver vitamin E levels. Furthermore, correlation analysis provided additional insights into the relationships between vitamin E and key antioxidant parameters in the livers of bats. On the other hand, no correlation was observed between vitamin E and key antioxidant parameters in the heart. Intriguingly, vitamin E was not detected in the kidneys, likely due to physiological factors and the prioritization of vitamin E mobilization in the heart, where it serves critical physiological functions. This unexpected absence of vitamin E in bat kidneys highlights the unique metabolic demands and prioritization of vitamin mobilization in wild animals like bats, compared to conventional animal models. These findings provide insight into the intricate distribution and utilization of vitamin E in bats, emphasizing the influence of dietary intake and metabolic adaptations on vitamin E levels in different organs.

## 1. Introduction

Vitamins play a crucial role in the health and physiological functions of all organisms, including bats. However, despite their significance, there is a scarcity of studies focusing on vitamin levels in bat species. Understanding the vitamin requirements and distribution in bats is essential for comprehending their dietary needs, metabolic processes, growth, and overall health. In addition, vitamin deficiencies may have adverse effects on bat health [1,2].

Due to their unique physiology and behavior, bats have developed specialized dietary preferences, which can vary widely among different species [3]. These dietary variations may significantly influence the intake and distribution of essential vitamins, such as vitamin E, within bat populations. Given the pivotal role of vitamin E in antioxidant defense and immune function, understanding its levels in bats is crucial for comprehending their physiological adaptations and health. As a key player in the defense against oxidative stress, vitamin E becomes paramount in sustaining the health and adaptability of bat populations. The varying levels of vitamin E across different bat species not only reflect their dietary distinctions but also shed light on the nuanced adaptations these creatures have developed over time. Therefore, addressing the existing knowledge gap in this area is imperative to gain insights into the intricate relationship between bat ecology, dietary habits, and vitamin E metabolism.

A recent study examining the amounts of vitamin C in several bat species’ organs from different dietary groups revealed fascinating insights into their dietary habits [4]. The study found that vitamin C levels in bat organs varied significantly, with the kidneys showing the lowest quantities and the heart, liver, and brain showing the greatest concentrations on average. Regarding vitamin E, no information is available so far. A comparative analysis of amounts of vitamin E in the liver and skeletal muscle has only been performed in one study involving several bat species during hibernation and summer activity [5].

In this work, we analyzed the levels of vitamin E in the heart, liver, and kidneys of four species of adult male bats, each representing a different feeding group. The bat species selected for this study were collected in southern Brazil, allowing us to examine vitamin E variations in bats from a specific geographical region. Understanding the distribution of vitamin E across these vital organs can provide insights into the metabolic homeostasis of bats, irrespective of their dietary preferences.

Through this investigation, we hope to contribute to the missing knowledge surrounding the dietary habits and nutritional requirements of bats, with implications for their conservation and management. By identifying potential disparities in vitamin E levels across different feeding groups and organs, we seek to reveal valuable insights into the remarkable adaptability and metabolic balance of bats. This knowledge not only enhances our understanding of bat biology but also holds implications for conservation efforts and ecosystem dynamics. Therefore, the exploration of vitamin E in bats stands as a crucial avenue for unraveling the intricacies of their dietary ecology and its broader implications for their health and ecological roles.

## 2. Material and Methods

### 2.1. Ethical Aspects

The capture of bats for this study was conducted in compliance with regulations and ethical standards. Specifically, the research was carried out under the authorization of the CONCEA (No. 33339, the National Council for the Control of Animal Experimentation), and SISBIO (No. 47202-1, the Brazilian Biodiversity Information and Authorization System). Furthermore, the study received approval from the University’s Ethics Committee on the Use of Animals (No. 28645).

### 2.2. Animals and Sample Collection

Table 1 provides the scientific name of each bat species, their feeding habits, and the season of sample collection. The capturing process, organ removal, and storage details were previously documented [6]. In summary, from summer 2018 to winter 2019, 39 adult male bats were captured in southern Brazil. The degree of ossification in wing elements was used to distinguish adult bats, following established techniques [7]. Different techniques for capture were used according to the type of shelter. To guarantee fasting, the bats were captured early in the evening and immediately put to death with an intraperitoneal injection of ketamine (60 mg/kg) and xylazine (10 mg/kg). It was not possible to obtain blood samples because of the euthanasia procedure. After euthanasia, they were immediately frozen in plastic bags with liquid nitrogen and placed on dry ice until they were taken to the local facility where they were stored at −80 °C.

### 2.3. Organ Processing

The heart, liver, and kidneys underwent manual maceration in a solution comprising potassium chloride (120 mmol/L), desferroxamine (1.5 mmol/L), phenylmethyl-sulfonyl fluoride (0.201 mmol/L), and 10 mL K3PO4 buffer (30 mmol/L). The samples were then centrifuged twice for ten minutes at 1700× *g* after being sonicated three times for ten seconds each. Following that, the resultant supernatant was divided into 1.5 mL microtubes and kept cold at −80 °C. Before each assay, centrifugation at 14,000× *g* was performed for 5 min.

### 2.4. Vitamin E Extraction and Assay

The extraction of Vitamin E (α-tocopherol) followed the method described by Barbas et al. 1997 [8]. Briefly, 50 μL of the previously frozen aliquots were used and mixed with 2 mL of n-hexane. This mixture was evaporated in a centrifugal concentrator and the residue redissolved in 200 μL of chloroform/methanol (1:3) for subsequent use in the HPLC assay. Using an HPLC system (Agilent Technologies, Santa Clara, CA, USA) and a 15 cm × 4.6 mm column (Nucleosil 120 C-18) with a constant flow rate of 2 mL per minute, vitamin E levels were determined. The water/methanol ratio was 93.5:3.5 (*v*/*v*). Fluorescence (295 nm excitation and 350 nm emission) was used for detection. The vitamin E had a 5 min retention period. The expression for vitamin E levels was nmol of vitamin E/mg protein [8].

### 2.5. Statistical Analysis

We used a one-way PERMANOVA to examine whether there were any significant variations in the amounts of vitamin E in each organ across the four bat species. Using the Shapiro–Wilk test to first determine normality, we next used Levene’s test to confirm the homogeneity of variances. We used the non-parametric test One-way PERMANOVA followed by a Bonferroni-corrected PERMANOVA pairwise comparison to examine pairwise differences across the bat species because these assumptions were not fulfilled (<0.05).

### 2.6. Correlation

We compared the levels of vitamin E measured in this study with the levels of different antioxidant enzymes and oxidative stress marker levels measured in our previous studies [4,6]. We calculated Spearman correlation coefficients for pairs of variables and tested their significance using two-tailed tests.

## 3. Results

### 3.1. Vitamin E Levels

Vitamin E levels measured in the heart and liver of the four bats are presented in Figure 1. In the heart, no significant differences were found. In the liver, significantly higher levels of vitamin E were observed in nectarivorous and frugivorous while lower levels were observed in hematophagous bats. No trace of vitamin E was detected in the kidneys. When comparing the levels of vitamin E between the heart and liver, significant differences were observed for nectarivorous (corrected *p* = 0.0009), frugivorous (corrected *p* = 0.0007), and insectivorous bats (corrected *p* = 0.0019). Comparing the amounts of vitamin E in the hematophagous bats’ liver and heart, no differences were found (corrected *p* = 0.94).

### 3.2. Correlation

The correlation between VitE measured in this study and other biochemical parameters (oxidative markers antioxidants) measured in our previous studies [4,6] are presented in Table 2 (for the heart) and Table 3 (for the liver).

In the heart, no correlation was observed between vitamin E and the biochemical parameters (Table 2). In the liver, vitamin E was positively correlated with GSSG/GSH, SOD, and GPx and negatively associated with MDA, fumarase, and GST (Table 3). No correlation was observed between vitamin E and Carbonyl, NO_2_ and NO_3_, vitamin C, and H_2_O_2_ consumption.

## 4. Discussion

In this study, vitamin E levels in the heart, liver, and kidneys of four species of adult male bats from various dietary groups were assessed and compared. Details on each bat species may be found in our previous work [6].

Our study found that there were no significant differences in vitamin E levels in the heart among the four bat species, indicating that vitamin E levels in the heart may be maintained through homeostatic mechanisms. The heart has a high metabolic rate and therefore requires a constant supply of antioxidants, regardless of the feeding group. On the other hand, higher levels of vitamin E in the liver were observed in nectarivorous and frugivorous bats, while lower levels were observed in hematophagous bats. Nectarivorous and frugivorous bats consume diets that include antioxidants like vitamin E, which may explain the higher levels observed in their livers.

Nevertheless, while nectar and fruits contain antioxidants, including vitamin E, the overall concentration is lower compared to other dietary sources. Investigations into nutritional composition and bioactive compounds in fruits, like mango pulp, revealed lower levels of vitamin E in mature fruit [9]. While nectarivorous and frugivorous bats may not consume high levels of vitamin E in their routine meals, alternative interpretations propose that these species could have developed enhanced liver or tissue storage capacities for vitamin E. This adaptation may allow them to efficiently store and utilize vitamin E even when dietary intake is limited [10]. Furthermore, these species might rely on alternative primary antioxidants, showcasing diverse evolutionary strategies to combat oxidative stress [11].

In addition, the variability in nutritional profiles among different species within the same feeding guild, as highlighted by the study conducted by Dierenfeld and Seyjaget (2000) [12], should also be taken into account. The comparison of vitamin E status in blood samples from three different species of fruit bats, all supposedly consuming the same diet, revealed significant differences across these species. The observed differences in nutritional status among the fruit bats suggest that factors such as feeding behavior, digestive physiology, metabolic rates, and nutrient utilization may contribute to variations in nutrient levels, including vitamin E, even when animals are exposed to similar dietary sources. This insight emphasizes the need for a nuanced understanding of dietary habits and their effects on nutrient intake and utilization in different species, particularly within the context of conservation and captive animal management.

In contrast, hematophagous bats feed on blood, which is low in antioxidants and may therefore have lower vitamin E levels. It is interesting to note that there was no discernible variation in the amounts of vitamin E found in the liver and heart of hematophagous bats. This finding suggests that hematophagous bats might have a more uniform distribution of vitamin E between their heart and liver.

Vitamin E is synthesized only by plants and algae, making it essential for all animals to obtain it from their diet [13]. Once ingested, this fat-soluble vitamin is stored in the liver and released to other organs as needed [14]. Research in pigs has shown that the liver has the highest storage capacity for vitamin E, with the concentration of this vitamin being significantly higher in the liver after a diet highly supplemented with vitamin E [15]. The concentration of vitamin E in the liver can decrease significantly after a period of depletion, while the concentrations in adipose tissue and skeletal muscle remain relatively stable during short-term depletion [16]. Vitamin E levels in serum and liver appear to be correlated within a certain range in growing pigs, indicating that the serum levels reflect the immediate nutritional status of the animal, while the liver stores represent its long-term nutritional history [17]. Moreover, the liver plays a central role in the uptake, distribution, metabolism, and storage of vitamin E, emphasizing its significance in maintaining vitamin E homeostasis in the body [18]. The high amount of vitamin E obtained by nectarivorous and frugivorous bats is then stored in the liver and continuously released to support vital functions, such as heart health. Vitamin E has also been detected in insects [19], which explains its intermediate concentration in the liver of insectivorous bats.

The distribution pattern of vitamin E in the heart and liver of hematophagous bats suggests that, due to their low dietary intake of this vitamin, it is rapidly mobilized to the heart. The similarity in vitamin E distribution between these organs indicates efficient transport mechanisms that prioritize the heart’s access to this essential nutrient. Hematophagous bats, which primarily feed on blood, may face challenges in obtaining sufficient vitamin E from their diet. However, their ability to swiftly allocate the available vitamin E to the heart highlights their adaptability and metabolic efficiency. This adaptation may be critical for meeting the high energy demands of the heart during flight and other physiological activities.

The correlation analysis provided additional insights into the relationships between vitamin E and key antioxidant parameters in the livers of bats.

In the liver, where many metabolic processes take place, the levels of vitamin E can influence the activity of key antioxidant enzymes and the balance of oxidized and reduced glutathione. The interconnection between vitamin E and glutathione (GSH) is notable. Recent evidence suggests that vitamin E enhances GSH levels [20], fostering a robust redox balance. GSH, a vital cellular antioxidant, forms a potent redox pair with its oxidized form (GSSG). The elevation of vitamin E contributes to a strengthened GSH antioxidant system [21].

Furthermore, the role of vitamin E extends to enzymatic antioxidants, such as Superoxide Dismutase (SOD). Vitamin E’s antioxidant properties mitigate oxidative stress by scavenging free radicals [22]. This, in turn, supports SOD in its role as an enzymatic antioxidant, creating a synergistic relationship.

The relationship between vitamin E and Glutathione Peroxidase (GPx) is crucial for the liver’s antioxidant capacity. Vitamin E prevents the depletion of glutathione, a co-factor for GPx [23]. Higher vitamin E levels support efficient GPx function, reinforcing the antioxidant defense mechanism against reactive oxygen species.

In summary, when the levels of vitamin E are higher in the liver, it can contribute to a more efficient antioxidant defense system by positively influencing the activity of SOD and GPx. Additionally, vitamin E can help maintain the balance between oxidized glutathione (GSSG) and reduced glutathione (GSH) in favor of reduced glutathione. This is important because glutathione is a major antioxidant in cells and the ratio of GSSG/GSH can serve as an indicator of oxidative stress.

On the other hand, the levels of vitamin E in the liver were negatively correlated with some biochemical parameters such as malondialdehyde (MDA), fumarase, and glutathione S-transferase (GST).

Malondialdehyde (MDA) is a marker of lipid peroxidation, which is a process associated with oxidative damage to cell membranes. Higher levels of MDA indicate increased lipid peroxidation and oxidative stress. A negative correlation between VitE and MDA would suggest that higher levels of VitE are associated with lower levels of lipid peroxidation in the liver [24], indicating a potential protective effect against oxidative damage.

Fumarase is an enzyme involved in the tricarboxylic acid (TCA) cycle, also known as the citric acid cycle or Krebs cycle, which plays a central role in cellular energy production. A negative correlation between VitE and fumarase levels could indicate a potential influence of VitE on the TCA cycle and cellular energy metabolism in the liver of bats [25].

Glutathione S-transferase (GST) is an important enzyme involved in the detoxification of reactive electrophiles and peroxides, contributing to cellular defense against oxidative stress. A negative correlation between VitE and GST levels may suggest a potential regulatory role of VitE in the liver’s antioxidant defense mechanisms, influencing the activity of GST and cellular detoxification processes [26].

These negative correlations between VitE levels and MDA, fumarase, and GST in the liver of bats could imply that VitE plays a role in mitigating oxidative stress, influencing cellular energy metabolism, and modulating antioxidant defense mechanisms in bat liver tissue. Further research is needed to validate these relationships and understand the underlying mechanisms governing these interactions in bat liver metabolism.

The lack of correlation between vitamin E levels and associated parameters in the heart may be attributed to the unique physiological dynamics of the cardiac environment. The heart is a vital organ responsible for pumping blood throughout the body, and its constant rhythmic activity ensures a steady flow of blood and oxygen delivery to tissues. Unlike the liver, which is involved in various metabolic processes and is exposed to fluctuations in oxidative stress and metabolic demands, the heart’s function is more focused on maintaining a consistent supply of oxygen and nutrients to support its continuous contractile activity.

The relatively constant and rhythmic nature of the heart’s function may lead to a more stable environment compared to the liver, where metabolic activities can fluctuate. This relatively stable environment in the heart may contribute to a consistent and less variable status of antioxidant parameters, despite their high levels. As a result, the influence of vitamin E levels on these parameters may not be as pronounced or detectable in the heart compared to the liver.

Additionally, the unique microenvironment and cellular composition of the heart tissue, which is primarily composed of cardiac muscle cells (cardiomyocytes), may also contribute to the observed lack of correlation. The specialized structure and function of cardiomyocytes, along with the specific distribution of antioxidant defense systems within the heart tissue, could result in different responses to changes in vitamin E levels compared to the liver. Further research into the specific mechanisms governing antioxidant responses in the heart tissue of bats may provide additional insights into this observation.

The lack of a significant correlation between vitamin E and vitamin C levels in the heart and liver may be due to differences in the way that these antioxidants function in the body. While both vitamin E and vitamin C are important antioxidants that protect cells from oxidative damage, they have different mechanisms of action and may be more effective in different parts of the body. For example, vitamin E is a lipid-soluble antioxidant that is primarily located in cell membranes, where it protects against lipid peroxidation. In contrast, vitamin C is a water-soluble antioxidant that is found in both intracellular and extracellular fluids, where it scavenges free radicals and regenerates other antioxidants. Therefore, it is possible that the levels of vitamin E and vitamin C do not correlate because they are performing different functions in the heart and liver.

Nevertheless, it is important to highlight that vitamin C metabolism is intimately associated with vitamin E metabolism in different species [27,28]. As such, the variable metabolic cycling of vitamin C across species, as established in our earlier work [4], may influence the interpretation of vitamin E dynamics, especially in nectarivorous and frugivorous species. In these groups, vitamin C may predominantly serve as the primary antioxidant, potentially sparing hepatic vitamin E depletion due to its distinct metabolic role. The reported differences in vitamin C metabolism across species could contribute to variations in the impact on hepatic vitamin E depletion. Specifically, the nectarivorous and frugivorous species may experience slower vitamin E depletion from liver tissues compared to other species. This emphasizes the need to consider the interplay between vitamin C and E metabolism in our interpretations and discussions, shedding light on the nuanced antioxidant roles within different ecological niches.

An intriguing finding of this study is that vitamin E was not detected in the kidneys, indicating an absence or low amount of the vitamin in this organ (below the technique’s limit of detection). Given that the same method detected significant levels in the heart and liver, we can infer that the technique was not the issue. The absence of vitamin E in the kidneys is likely due to physiological factors. After dietary intake, Vitamin E is subject to hepatic metabolism, and its metabolites are then incorporated into lipoproteins for distribution to various tissues [29]. Due to the significant energy expenditure during flight, bats prioritize the mobilization of vitamin E in the heart due to its high metabolic rate, where this vitamin may exert antioxidant activity and other physiological functions [30]. Consequently, the relatively low significance of vitamin E transport to bat kidneys may explain the absence of this vitamin in the organ. This finding is unexpected as vitamin E is typically found in the kidneys of many animal models [31,32]. However, it is important to consider that the physiological demands and priorities for vitamin mobilization may differ significantly between animal models and wild animals, particularly in the case of bats, which have unique metabolic requirements due to their flying behavior.

## 5. Conclusions

In this study, we investigated the vitamin E levels in the heart, liver, and kidneys of four species of adult male bats with distinct dietary patterns. Our findings revealed that vitamin E levels in the heart were consistent across all four bat species, suggesting the presence of homeostatic mechanisms maintaining these levels. Notably, the liver exhibited significantly higher vitamin E levels in nectarivorous and frugivorous bats, while hematophagous bats displayed lower levels, indicating a correlation between dietary intake and liver vitamin E levels. Furthermore, correlation analysis provided additional insights into the relationships between vitamin E and key antioxidant parameters in the liver of bats. On the other hand, no correlation was observed between vitamin E and key antioxidant parameters in the heart. Surprisingly, vitamin E was not detected in the kidneys, likely due to physiological factors and the prioritization of vitamin E mobilization in the heart, where it serves crucial physiological functions. This unexpected absence of vitamin E in bat kidneys underscores the unique metabolic requirements and priorities for vitamin mobilization in wild animals, such as bats, compared to traditional animal models. These findings shed light on the intricate distribution and utilization of vitamin E in bats, emphasizing the significance of dietary intake and metabolic adaptations in shaping vitamin E levels across different organs.

## Figures and Tables

**Figure 1 life-14-00266-f001:**
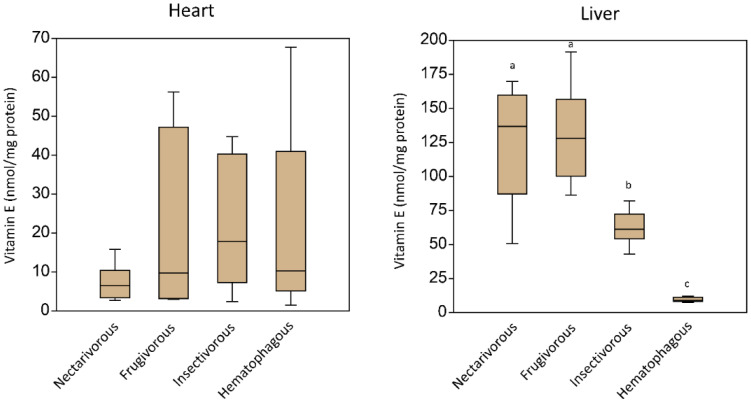
Vitamin E levels measured in the heart and liver of the four adult male bat species. Data are presented as the median (interquartile range). Variations in letters signify statistical differences between different species. While similar letters imply no statistical differences (*p* > 0.05), distinct letters show significant differences among species (*p* < 0.05).

**Table 1 life-14-00266-t001:** Bats species and their feeding habits, along with the number (*n*) of specimens captured and the respective capture season.

Bat Species	Feeding Habit	*n*	Season
*Glossophaga soricina*	Nectarivorous	10	Autumn (2019)
*Sturnira lilium*	Frugivorous	10	Winter (2019)
*Molossus molossus*	Insectivorous	10	Summer (2018)
*Desmodus rotundus*	Hematophagous	9	Summer (2018)

**Table 2 life-14-00266-t002:** Spearman’s correlation between vitamin E and other biochemical parameters measured in the heart. No statistical difference (*p* < 0.05) was observed.

Parameters	r	*p*
Carbonyl	0.03	0.9838
MDA	−0.13	0.4790
NO_2_ and NO_3_	−0.11	0.5687
GSSG/GSH	0.04	0.8608
VitC	−0.24	0.2425
H_2_O_2_↓	−0.09	0.6444
SOD	−0.01	0.9764
Fumarase	0.11	0.5529
GPx	−0.19	0.3219
GST	−0.25	0.1793

**Table 3 life-14-00266-t003:** Spearman’s correlation between vitamin E and other biochemical parameters measured in the liver. The asterisk indicates statistical significance (*p* < 0.05).

Parameters	r	*p*
Carbonyl	−0.32	0.0943
MDA	−0.65	0.0004 *
NO_2_ and NO_3_	−0.16	0.4146
GSSG/GSH	0.48	0.0490
VitC	0.19	0.3444 *
H_2_O_2_↓	0.15	0.4412
SOD	0.72	0.0001 *
Fumarase	−0.67	0.0001 *
GPx	0.42	0.0259 *
GST	−0.75	0.0001 *

## Data Availability

The data presented in this study are available in Section 3.

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
