# Peer review of "Distribution and Utilization of Vitamin E in Different Organs of Wild Bats from Different Food Groups"

_life, 2024, doi:10.3390/life14020266_

Round 1
Reviewer 1 Report
Comments and Suggestions for Authors
The paper may be published after moderate revision even if the species considered may not be of particular scientific interest
Comments on the Quality of English LanguageModerate editing of English language required
Author Response
We thank the reviewer for taking the time to review our manuscript.
Reviewer 2 Report
Comments and Suggestions for Authors
2.4. Vitamin E Extraction and Assay
You must declare the manufacturer, city, country for HPLC system
Why do you consider that the levels of vitamin E can be expressed as nmol vit E / mg protein? Did you determine the vitamin E from extracted protein? Why not mg tissue if you extract the the vitamin E from „frozen aliquotes”
2.5. Statistical Analysis
You have to declar the level of significance
3. Results
In the heart results, the range of values per group is huge and the average value is not representing the data string. From this reason, there are not statistical differences between groups. The results are irrelevant if the data strings are not homogenous.
Overall, the manuscript contains only one parameter (vitamin E) determined in 3 types of samples, but results declared for 2 organs. I consider that the results must be improved with other studied parameters so the scientific soundness to be reached
Author Response
Reply: We thank the reviewer for taking the time to review our manuscript. We have provided below a point-to-point reply to them all, highlighting all changes in red in the revised manuscript.
You must declare the manufacturer, city, country for HPLC system
Reply: Done
Why do you consider that the levels of vitamin E can be expressed as nmol vit E / mg protein? Did you determine the vitamin E from extracted protein? Why not mg tissue if you extract the vitamin E from „frozen aliquotes”
Reply: The levels of vitamin E are often expressed as nmol of vitamin E per milligram of protein because this unit of measurement allows for standardization and comparison across different samples. By expressing the concentration of vitamin E in relation to the amount of protein present, it provides a normalized value that can be used to compare levels of vitamin E across different biological samples or experimental conditions. This standardization helps in making meaningful comparisons and drawing accurate conclusions from the data. As such, it has been used in our previous studies (e.g. https://www.mdpi.com/2075-1729/12/12/2121), which allowed a direct comparison with the current study. Please see lines 123-125 and a proper discussion in lines 197-217.
2.5. Statistical Analysis
You have to declar the level of significance
Reply: Done
- Results
In the heart results, the range of values per group is huge and the average value is not representing the data string. From this reason, there are not statistical differences between groups. The results are irrelevant if the data strings are not homogenous.
Reply: The variability in the range of values within each group does not necessarily invalidate the biological significance of our results. Even though there is a wide range of values, it's important to consider the biological context of analyzing biochemical parameters from wild animals. For this reason, despite the wide range, our results have biological relevance and validity.
Overall, the manuscript contains only one parameter (vitamin E) determined in 3 types of samples, but results declared for 2 organs. I consider that the results must be improved with other studied parameters so the scientific soundness to be reached.
Reply: The analysis of VitE per se is scientifically sound as we have properly discussed (i.e. with a focus on the distribution and use of VitE, not including other parameters), similar to our previous work focused on VitC https://www.mdpi.com/2075-1729/12/12/2121. Any additional parameter would just be complementary, not a necessity to reach scientific soundness. But to complement this work, we have correlated and discussed the results of VitE and VitC (from our previous work), especially because vitamin C metabolism is intimately associated with vitamin E metabolism in different species. Please see lines 123-125 and a proper discussion in lines 197-217.
Reviewer 3 Report
Comments and Suggestions for Authors
Interesting comparative study, and cleanly presented. Just curious, was there an underlying motivation (disease, overall health status?) or rather a basic knowledge quest driving this scientific query? In line 35 you suggest there may be underlying vitamin deficiency issues but the citation refers to vitamin C problems, and you missed an important paper on vitamin E deficiency in bats (Heard et al., 1996; J Zoo Wildl Med) - which actually also includes data on vitamin E levels in plasma, liver, heart, and kidney tissues. So those absolutely need to be converted to the same units you used and discussed, especially since the diets of the bats in the 1996 study were well defined.
Without access to some of the earlier work by your research group on vitamin C across chiropteran species (referred to in line 44), it is difficult to follow some of the full interpretations of the data presented here. Vitamin C status and metabolism is intimately associated with vitamin E metabolism in other species, and need to be discussed as such, even if speculatively. We know that some - not all, to current knowledge - bat species require vitamin C be provided by the diet; presumably those species (i.e. frugivores for sure) that ecologically consume foodstuffs high in vitamin C never evolved the mechanisms for synthesis of ascorbic acid. That physiology may not be the same across other feeding categories - and this may influence vitamin E metabolism and storage as well. This needs to be considered in your interpretations.
In line 134, however, you state that nectar and fruits can be good dietary sources of antioxidants, including vitamin E. That latter part, however is actually not true - those feed ingredient categories are notoriously low in vitamin E across all samples examined, and species that evolved with those feeding habits have considerably lower circulating plasma or serum levels than folivorous, granivorous or other herbivorous feeding guilds. Omnivores are higher, and those feeding on animal-based diets (including blood as an animal-based product and insectivores) would typically consume higher vitamin E levels than the other 2 groups (nectar or fruit feeders) - this has been documented within other avian as well as mammalian groups. [Admittedly, however, the blood concentration will vary depending upon the prey species.] Suggest alternative interpretations in that species that don't necessarily consume high levels of vitamin E in routine meals (nectarivores or frugivores) may have a better liver / tissue storage capacity, or simply utilize other primary antioxidants. OR - behaviorally those bats were able to consume a meal prior to capture and your Desmodus perhaps were not able to do so, with the insectivores intermediate? Seasonality may play a role here as well, as availability or composition of food items may have varied. Did you analyze any food items throughout your work with these various species?
Another paper (with captive bats) that may be of interest is Dierenfeld and Seyjaget 2000 (J Zoo Wildlife Med) who compared vitamin E (and other nutrients) status in blood samples from 3 different species of fruit bats all consuming (in theory) the same diet, and significant differences were seen across these species. So even within a feeding guild you may have habits/selectivity that alter the nutritional profiles from different tissues. This possibility should be acknowledged as well.
Lines 157-158, perhaps add the word "bats" along with the adjectives insectivorous, nectarivorous and frugivorous or change the words to nouns (insectivores, nectarivores, frugivores).. Here you also state that the insectivorous bat heart and liver levels are higher than nectarivorous or folivorous bat levels measured. This statement is not true. There were no differences among heart concentrations, and in the figure, the insectivorous bat liver levels are signficantly lower than the first 2 feeding categories. This must be corrected.
With your previous work on vitamin C in this group, suggest to emphasize that variable metabolic cycling with vitamin C across species may also come into play in your vitamin E interpretations/discussion, with vitamin C fulfilling more tje primary antioxidant role vs vitamin E for the nectarivorous and frugivorous species - hence not depleting vitamin E from liver tissues as rapidly. In other words, the vitamin C metabolic differences across your species (reported in your earlier work?) may have impacted or "spared" hepatic vitamin E depletion in some but not all species.
The lack of vitamin E in your kidney tissues is interesting, especially since concentrations of this nutrient have been reported for other bat species (Heard et al. 1996). Would love for you to convert data from the Heard et al. paper (or vice versa, from yours to their units) and compare the vitamin E levels you did measure in cardiac and hepatic tissue as part of your Discussion section.
Author Response
Reply: We thank the reviewer for the meticulous revision of our manuscript and the interesting suggestion that helped us to better present our work. We have provided below a point-to-point reply to them all, highlighting all changes in red in the revised manuscript.
Interesting comparative study, and cleanly presented. Just curious, was there an underlying motivation (disease, overall health status?) or rather a basic knowledge quest driving this scientific query? In line 35 you suggest there may be underlying vitamin deficiency issues but the citation refers to vitamin C problems, and you missed an important paper on vitamin E deficiency in bats (Heard et al., 1996; J Zoo Wildl Med) - which actually also includes data on vitamin E levels in plasma, liver, heart, and kidney tissues. So those absolutely need to be converted to the same units you used and discussed, especially since the diets of the bats in the 1996 study were well defined.
Reply: The motivation behind our study lies in a basic knowledge quest. The study aims to elucidate the variations in vitamin E distribution across organs in wild bats, shedding light on their physiological adaptations. Understanding how different bat species utilize and distribute vitamin E can contribute to our broader comprehension of their metabolic processes and ecological niches. While the study does not explicitly focus on disease or overall health status, it lays the groundwork for comprehending the nutritional aspects of these unique mammals in their natural habitat.
Regarding the Heard et al., 1996 study, it is indeed an interesting work on vitamin E deficiency in bats that we have missed and was now properly cited in the revised version [1]. However, converting those data and performing a direct comparison with our study is not feasible considering the significant number of differences between studies. Yes, it is easy to convert the data to the same units, but everything else would be a huge bias. The 1996 study had only a few individuals (three unaffected). It included both males and females, and the bat species studied is quite distant geographically and evolutionarily from ours. Additionally, different methods were used. As such, even a comparative discussion is not feasible.
Without access to some of the earlier work by your research group on vitamin C across chiropteran species (referred to in line 44), it is difficult to follow some of the full interpretations of the data presented here. Vitamin C status and metabolism is intimately associated with vitamin E metabolism in other species, and need to be discussed as such, even if speculatively. We know that some - not all, to current knowledge - bat species require vitamin C be provided by the diet; presumably those species (i.e. frugivores for sure) that ecologically consume foodstuffs high in vitamin C never evolved the mechanisms for synthesis of ascorbic acid. That physiology may not be the same across other feeding categories - and this may influence vitamin E metabolism and storage as well. This needs to be considered in your interpretations.
Reply: We have indeed correlated the levels of VitC and VitE but no significant result was observed. But we added this result (lines 123-125) and a proper discussion in the revised version (lines 197-206).
In line 134, however, you state that nectar and fruits can be good dietary sources of antioxidants, including vitamin E. That latter part, however is actually not true - those feed ingredient categories are notoriously low in vitamin E across all samples examined, and species that evolved with those feeding habits have considerably lower circulating plasma or serum levels than florivorous, granivorous or other herbivorous feeding guilds. Omnivores are higher, and those feeding on animal-based diets (including blood as an animal-based product and insectivores) would typically consume higher vitamin E levels than the other 2 groups (nectar or fruit feeders) - this has been documented within other avian as well as mammalian groups. [Admittedly, however, the blood concentration will vary depending upon the prey species.] Suggest alternative interpretations in that species that don't necessarily consume high levels of vitamin E in routine meals (nectarivores or frugivores) may have a better liver / tissue storage capacity, or simply utilize other primary antioxidants. OR - behaviorally those bats were able to consume a meal prior to capture and your Desmodus perhaps were not able to do so, with the insectivores intermediate? Seasonality may play a role here as well, as availability or composition of food items may have varied. Did you analyze any food items throughout your work with these various species?
Reply: Quite relevant comments that were specifically addressed in lines 144-153. Regarding the question, unfortunately, no food item was analyzed in our work.
Another paper (with captive bats) that may be of interest is Dierenfeld and Seyjaget 2000 (J Zoo Wildlife Med) who compared vitamin E (and other nutrients) status in blood samples from 3 different species of fruit bats all consuming (in theory) the same diet, and significant differences were seen across these species. So even within a feeding guild you may have habits/selectivity that alter the nutritional profiles from different tissues. This possibility should be acknowledged as well.
Reply: You make a valid point about the variability in nutritional profiles among different species within the same feeding guild, as highlighted by the study conducted by Dierenfeld and Seyjaget in 2000. The comparison of vitamin E status in blood samples from three different species of fruit bats, all supposedly consuming the same diet, revealed significant differences across these species. This finding underscores the importance of acknowledging the potential impact of species-specific dietary habits and selectivity on nutritional profiles within a feeding guild. We have addressed this issue in lines 154-164.
Lines 157-158, perhaps add the word "bats" along with the adjectives insectivorous, nectarivorous and frugivorous or change the words to nouns (insectivores, nectarivores, frugivores).. Here you also state that the insectivorous bat heart and liver levels are higher than nectarivorous or folivorous bat levels measured. This statement is not true. There were no differences among heart concentrations, and in the figure, the insectivorous bat liver levels are signficantly lower than the first 2 feeding categories. This must be corrected.
Reply: We have made the necessary adjustments as suggested. Please see lines 185-187.
With your previous work on vitamin C in this group, suggest to emphasize that variable metabolic cycling with vitamin C across species may also come into play in your vitamin E interpretations/discussion, with vitamin C fulfilling more the primary antioxidant role vs vitamin E for the nectarivorous and frugivorous species - hence not depleting vitamin E from liver tissues as rapidly. In other words, the vitamin C metabolic differences across your species (reported in your earlier work?) may have impacted or "spared" hepatic vitamin E depletion in some but not all species.
Reply: Excellent suggestions that were addressed in lines 207-218.
The lack of vitamin E in your kidney tissues is interesting, especially since concentrations of this nutrient have been reported for other bat species (Heard et al. 1996). Would love for you to convert data from the Heard et al. paper (or vice versa, from yours to their units) and compare the vitamin E levels you did measure in cardiac and hepatic tissue as part of your Discussion section.
Reply: As previously informed, such a direct comparison with our study is not feasible considering the significant number of differences between the studies.
Round 2
Reviewer 2 Report
Comments and Suggestions for Authors
Accept